# Experimental study on tailings cementation by MICP technique with immersion curing

**Changyu Jin*, Huiyang Liu, Mingxiao Guo, Yunfeng Wang, Jinyao Zhu**

Key Laboratory of Ministry of Education on Safe Mining of Deep Metal Mines, Northeastern University, Shenyang, Liaoning, 110004, China

\* jcy_snow@163.com

**Data Availability Statement:** All relevant data are within the article and its Supporting Information files.

**Funding:** This study was funded by the Fundamental Research Funds for Central

## Abstract

The filling mining method is an effective method for controlling ground stress and preventing surface subsidence in the mining field during exploitation of underground resources. Tailings can be utilized as the filling material, so as to realize the reuse of industrial waste. However, utilization of the traditional Portland cement as the cementing material for tailings leads to groundwater pollution. In addition, production of Portland cement results in consumption of a great amount of ore and air pollution. In this paper, a tailings cementation method by using the microbial induced calcite precipitation (MICP) technique with immersion curing is proposed. Tailings are cemented by the MICP technique with aerobic bacteria (*Sporosarcina pasteurii*) under a soaked curing environment. The variable control method is applied to investigate the factors influencing the cementation effects by the MICP technique with *Sporosarcina pasteurii*, including the bacterial solution concentration, the cementing solution concentration, the particle size of tailings, and the curing temperature. The results indicate that: when $OD_{600}$ of the *Sporosarcina pasteurii* solution is 1.6, the urea concentration in the cementing solution is 0.75 mol/L, the tailings are raw materials without grinding, and the curing temperature is 30˚C, the cementation effect is the best. In view of uneven calcification during MICP with *Sporosarcina pasteurii*, mixed *Sporosarcina pasteurii* and *Castellaniella denitrificans* are used for tailings cementation. Higher strength of cemented tailings is achieved. It is proved that the MICP technique with mixed aerobic bacteria and facultative anaerobes is an effective method for tailings cementation.

## 1. Introduction

Mineral resources are the necessities of human and economic development. With the continuous progress of society, the demand for mineral resources around the world continuously increases. The shallow resources are gradually exhausted, and open-pit mining gradually transits to underground mining [1,2]. Underground mining will inevitably lead to a large number of goaf in the strata, which is the main cause of rock instability and safety accidents. In addition, the decline of groundwater level, surface subsidence and tailings reservoir pollution are also troublesome problems in underground mining [3,4]. In order to solve the above problems, the filling mining method has become the preferred method for underground resources mining [5,6]. At present, Portland cement is commonly used as the cementing material for

Universities (N2101041) to CJ. The funders had no role in study design, data collection and analysis, decision to publish, or preparation of the manuscript.

**Competing interests:** The authors have declared that no competing interests exist.

tailings cementation. A great amount of Portland cement is filled underground, leading to high pH value and pollution of groundwater. Furthermore, carbon emission is inevitable in the process of cement production. The cement industry has become one of the main greenhouse gas emission industries [7]. Therefore, finding a tailings cementing material that can replace Portland cement has become a hot scientific research direction. The use of environmentally friendly tailings cementitious materials is of great significance for reducing carbon dioxide emissions and achieving carbon neutrality as soon as possible.

The microbial induced calcite precipitation (MICP) technique is a typical biomineralized consolidation method. In a certain physical and chemical environment, the free calcium ions in the solution are transformed into solid minerals dominated by calcium carbonate, so as to realize solidification of granular particles by controlling the biological organic matter. There are many kinds of solidification methods: urea hydrolysis, dentrification, ferric salt reduction, sulfate reduction, and photosynthesis etc., among which urea hydrolysis is widely used [8]. Urea hydrolysis refers to the continuous production of highly active urease by specific microorganisms (urease bacteria) in the process of metabolism. Urease will promote the hydrolysis of urea to generate carbon dioxide and ammonia, which combine with water to generate carbonate ions and ammonium ions, and increase the alkalinity of the environment. When calcium ions are added externally, calcium carbonate precipitates are formed in the environment. The reaction equation is shown in (1–6). The mechanism of the MICP technique based on urea hydrolysis is relatively simple and a large amount of calcium carbonate is precipitated within a short period. It is highly efficient, sustainable and pollution-free, and hence has attracted attention from many researchers around the world [9–11]. In 2004, Whiffin [12] first proposed cementation of sand particles by the MICP technique, so as to improve the strength and stiffness of sand. With properly selected bacillus, Dick et al. [13] observed calcium carbonate precipitation on the surface of weathered limestone by using the surface coating method, which greatly reduced the capillary water absorption coefficient of limestone surface. Tittelboom et al. [14] repaired concrete cracks by using the MICP technique. With silica as the protective carrier of bacteria, 0.3 mm wide cracks can be completely repaired. By using porous aggregates as the carrier for bacteria and lactic acid bacteria, Wicktor et al. [15] repaired cracks in concrete under immersion curing for 100 days. A great amount of white calcite precipitated in the cracks and the repair effect was good. Chu et al. [16] found a proportional relationship between the uniaxial compressive strength of the sand column cemented by the MICP technique and the amount of calcium carbonate precipitation, and proposed a linear empirical equation between them. Van Paassen et al. [17] conducted consolidation tests on a large volume of sand ($100m^3$) by using the MICP grouting. After 12 days of cyclic grouting, about half volume of sand was cemented, indicating the the MICP technique can enhance the integrity of sand. Deng et al. [18] mixed *Sporosarcina pasteurii* (*S. pasteurii*) and metal ore tailings directly and cured the mixture in a sealed environment. The effect of tailings cementation by the MICP was studied. Many scholars have conducted studies on repair and reinforcement of various materials, including concrete-based materials, soil and sand, by the MICP technique [11,19,20]. Surface brushing or grouting was adopted in most of the previous studies, which cannot effectively reinforce the interior of sand or soil mass.

$$CO(NH_2)_2 + H_2O \rightarrow CO_2 + 2NH_3 \qquad (1)$$

$$CO_2 + H_2O \leftrightarrow H_2CO_3 \qquad (2)$$

$$2NH_3 + 2H_2O \leftrightarrow 2NH_4^+ + 2OH^- \qquad (3)$$

**Table 1. Main chemical composition and content in tailings (wt%).**

| SiO$_2$ | Al$_2$O$_3$ | CaO | Fe$_2$O$_3$ | MgO | K$_2$O | P$_2$O$_5$ | MnO | SO$_3$ | TiO$_2$ | Na$_2$O |
|---|---|---|---|---|---|---|---|---|---|---|
| 77.57 | 13.60 | 2.89 | 2.73 | 2.40 | 0.48 | 0.11 | 0.08 | 0.07 | 0.04 | 0.03 |

$$2OH^- + H_2CO_3 \leftrightarrow CO_3^{2-} + 2H_2O \tag{4}$$

$$Ca^{2+} + Call \rightarrow Call - Ca^{2+} \tag{5}$$

$$Call - Ca^{2+} + CO_3^{2-} \rightarrow CaCO_3 \downarrow \tag{6}$$

To sum up, a tailings cementation method by mixed facultative anaerobes (*Castellaniella denitrificans* or *C. denitrificans*) and *Sporosarcina pasteurii* (*S. pasteurii*) is proposed in this paper. The influences of different mixing modes on the cementation effect are investigated so as to improve the cementation effect. Based on the research in this paper, it is found that in the process of soaking and curing, the cementation effect is better when facultative anaerobic bacteria and aerobic bacteria are used to act together on the tailings.

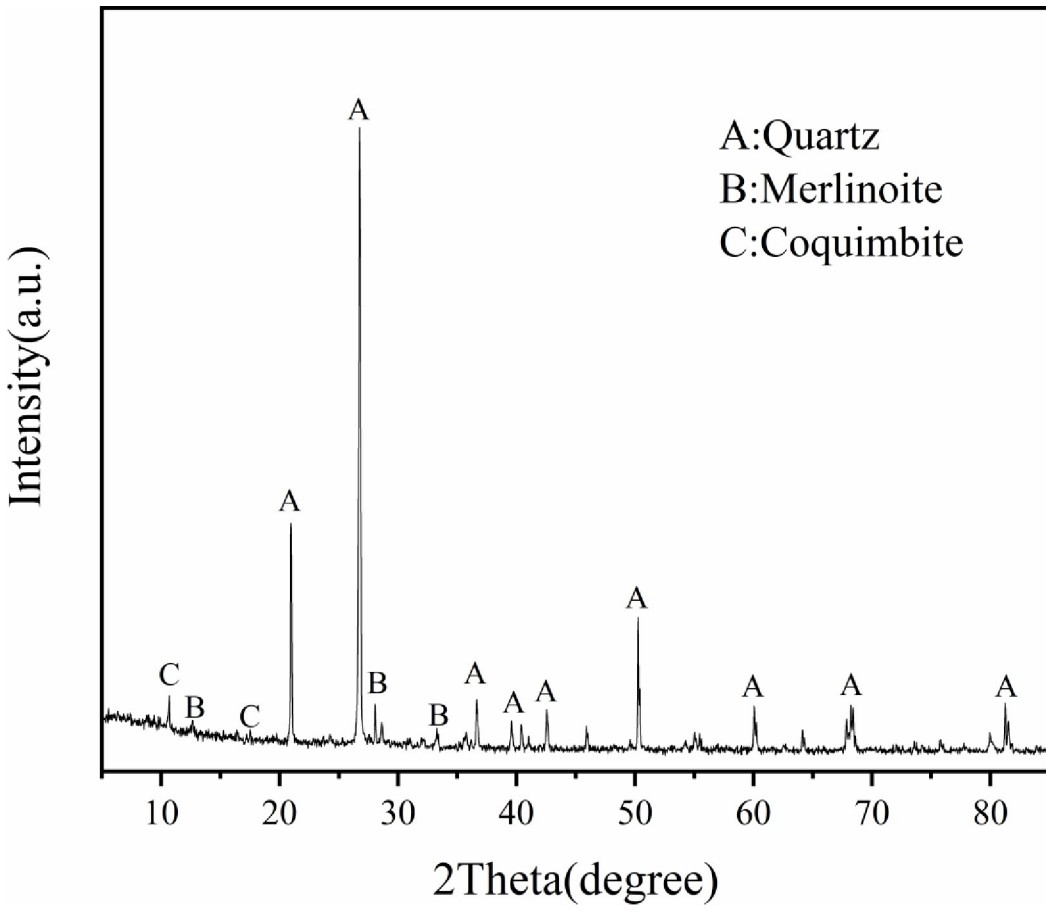

**Fig 1. XRD spectrum of tailings.**

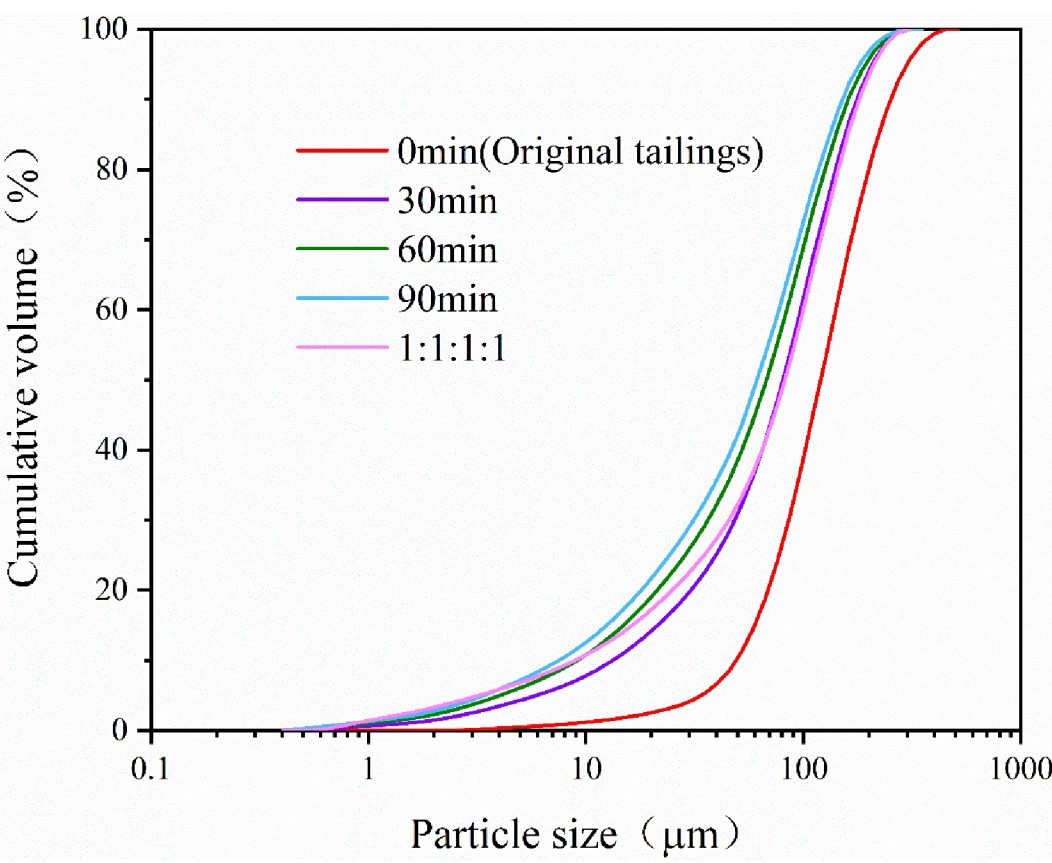

**Fig 2. Particle size distribution of tailings with different grinding time.**

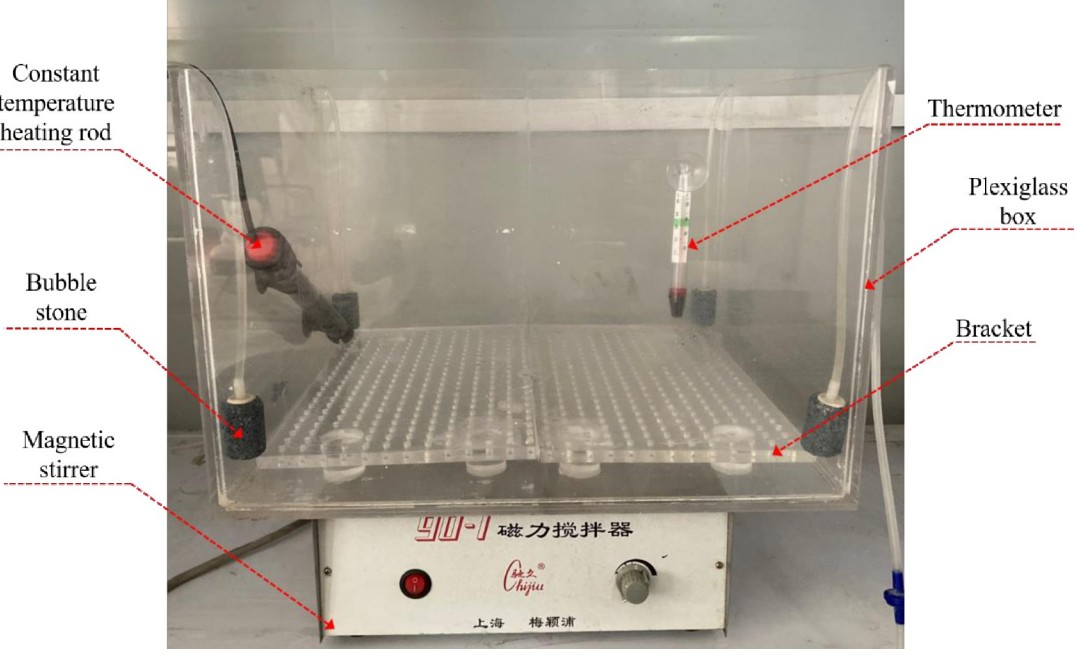

**Fig 3. Device for immersion curing.**

## 2. Test materials and device

### 2.1 Test materials

(1) **Tailings.** The tailings under study were collected from the tailings reservoir of an iron mine in Anshan, Liaoning, Northeast China. The main chemical composition of the tailings was determined by X-ray fluorescence analysis (XRF), which mainly include $SiO_2$ (77.57wt%), $Al_2O_3$ (13.59wt%), $CaO$ (2.89wt%), $Fe_2O_3$ (2.73wt%) and $MgO$ (2.40wt%), as well as some minor chemical components, as listed in Table 1. The crystal phase of tailings was determined by X-ray diffraction analysis (XRD). As shown in Fig 1, the main minerals are quartz, merlinoite and coquimbite. In order to investigate the effect of grain size on tailings cementation, the original tailings were ground by a vertical planetary ball mill (XQM-4L) for 30 min, 60 min and 90 min, respectively, with the composition of tailings remaining unchanged. The original tailings were mixed with the three types of ground tailings in a mass ratio of 1:1:1:1 to prepare tailings specimens with a new particle size distribution. A laser particle size analyzer (Mastersizer3000 manufactured by Malvern Panalytical) was employed to analyze the particle size distribution of five tailings specimens and the results are presented in Fig 2.

(2) **Microbes.** The bacteria used in this study were *S. pasteurii* and *C. denitrificans* obtained from China General Microbiological Culture Collection Center with a strain number

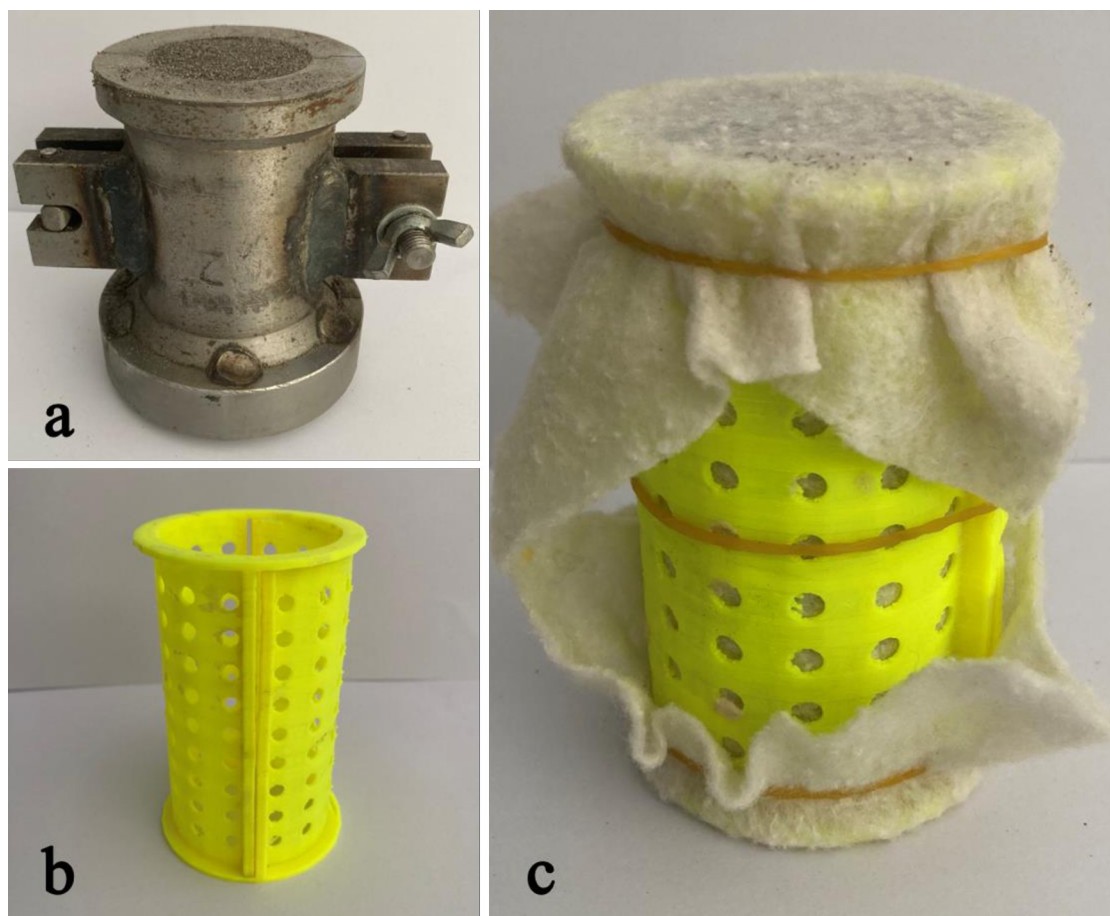

**Fig 4. Shaping and preparation of tailings specimen.**

of CGMCC 1.3687 and 1.10720, respectively. *S. pasteurii* is a kind of Gram-positive aerobic bacteria, while *C. denitrificans* is kind of Gram-negative facultative anaerobe. The microbes were cultured in nutrient medium containing 10 g/L peptone, 5 g/L NaCl, and 3 g/L beef extract. The pH value of the culture medium was adjusted to 7.0. The culture medium was sterilized in an autoclave before it was taken to the sterile bench for tests. The activated *C. denitrificans* and *S. pasteurii* were inoculated to the sterile culture medium, respectively, and put into a vibration incubator at 30˚C with an agitation speed of 140r/min.

**(3) Cementing solution.** The cementing solution mainly provides urea and $Ca^{2+}$ for microbial induced calcite precipitation during the immersion process, meanwhile, it provides necessary nutrients for the growth and reproduction of *S. pasteurii*. According to the previous studies on the MICP technique based on the immersion method [21], the main composition of the cementing solution includes urea, $CaCl_2$, $NH_4Cl$, $NaHCO_3$, nutrient broth and deionized water. Based on the reaction mechanism of calcium carbonate precipitation induced by urea hydrolysis of *S. pasteurii*, the concentration of urea and $CaCl_2$ remained unchanged for studies on the influence of cementing solution concentration on the cementation effect. The concentration of $NH_4Cl$, $NaHCO_3$ and nutrient broth were 0.28 mol/L, 0.025 mol/L and 5 g/L, respectively. As nitrate ions and acetate ions are required to participate in dentrification by *C. denitrificans*, an approximate amount of $Ca(NO_3)_2$ and $Ca(CH_3COO)_2$ was added into the cementing solution in addition to the above-mentioned substances when mixed *S. pasteurii* and *C. denitrificans* were used.

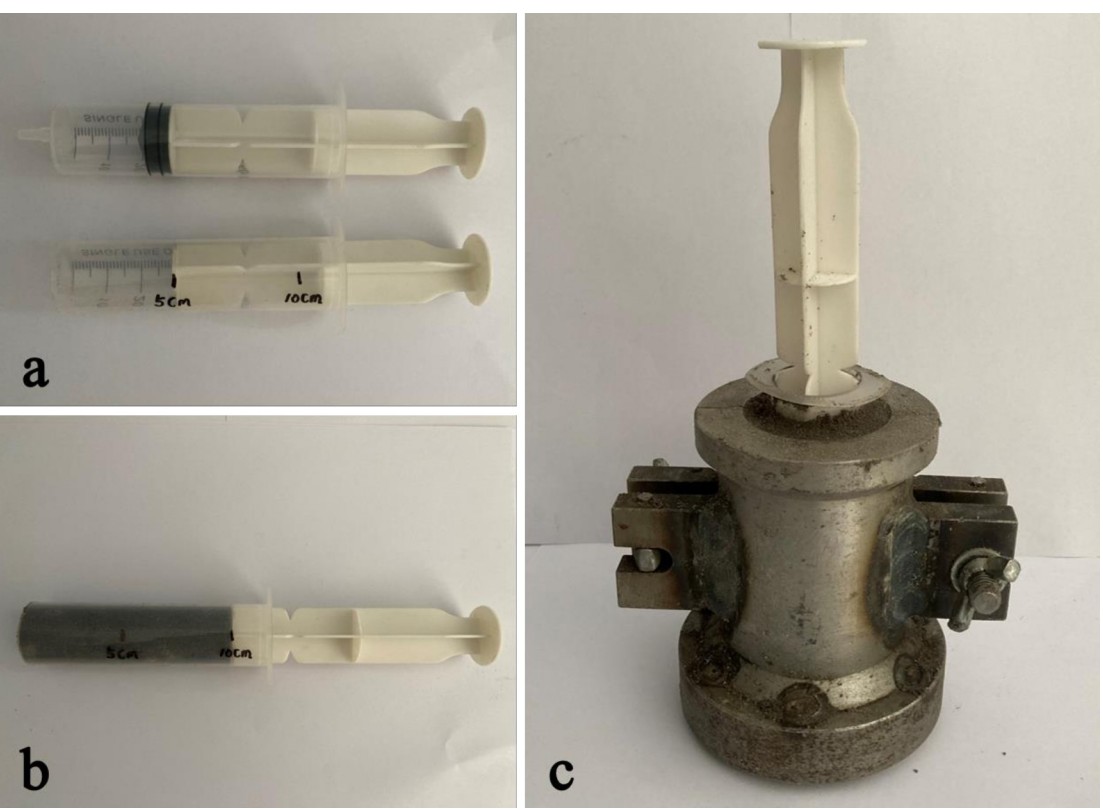

**Fig 5. Shaping of specimens with mixed bacteria.**

## 2.2 Test device

The device for immersion curing is shown in Fig 3, which mainly consists of a plexiglass box with a size of 450 mm×400 mm×300 mm for containing cementing solution and tailings specimen; a magnetic stirrer, which mixes the cementing solution in the plexigalss box evenly

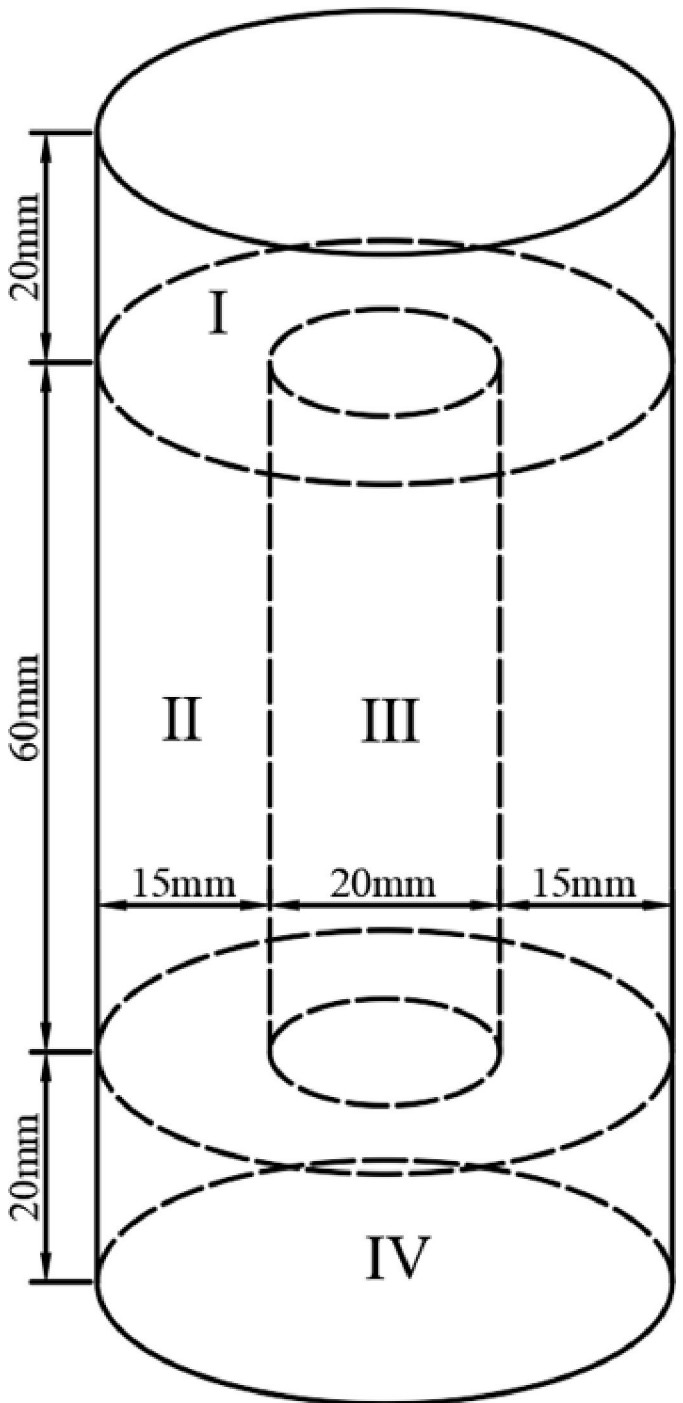

**Fig 6. Division of tailings specimen.**

by driving the magnetic rotor; a bracket with circular holes at equal intervals to facilitate full contact between the lower part of the specimen and the cementing solution; a thermostatic heating rod to maintain a constant temperature of the cementing solution; bubble stones to provide oxygen for the growth and reproduction of *S. pasteurii* in the plexiglass box; a thermometer for real-time display of the temperature and monitoring the temperature changes of the cementing solution.

## 3. Test method

### 3.1 Cementation method

Firstly, dry tailings and cultured *S. pasteurii* solution were mixed. Each tailings specimen was mixed with 70mL bacterial solution. The mixture was put into a steel mold (diameter 50 mm, height 100 mm, as shown in Fig 4A) for shaping and sample preparation. In order to avoid loss of tailings particles during the contact process between the cementing solution and the tailing specimen, the specimen was wrapped with 1 mm thick geotextile and laterally confined by a 3D printed mold (as shown in Fig 4B), so as to prevent specimen deformation during immersion curing. Finally, the upper and lower ends of the specimen were wrapped by geotextile (Fig 4C). 30 L cementing solution was injected into the plexiglass box, and the power supply of the magnetic stirrer and the thermostatic heating rod was connected. When a preset constant temperature was reached, the tailings specimen was put into the cementing solution and the air pump was turned on. After being immersed and cured under a constant temperature in an aerobic environment for 28 d, the specimen was taken out and dried for demolding. Tests were then conducted on the tailings specimen for analysis of the cementation effect. According to previous

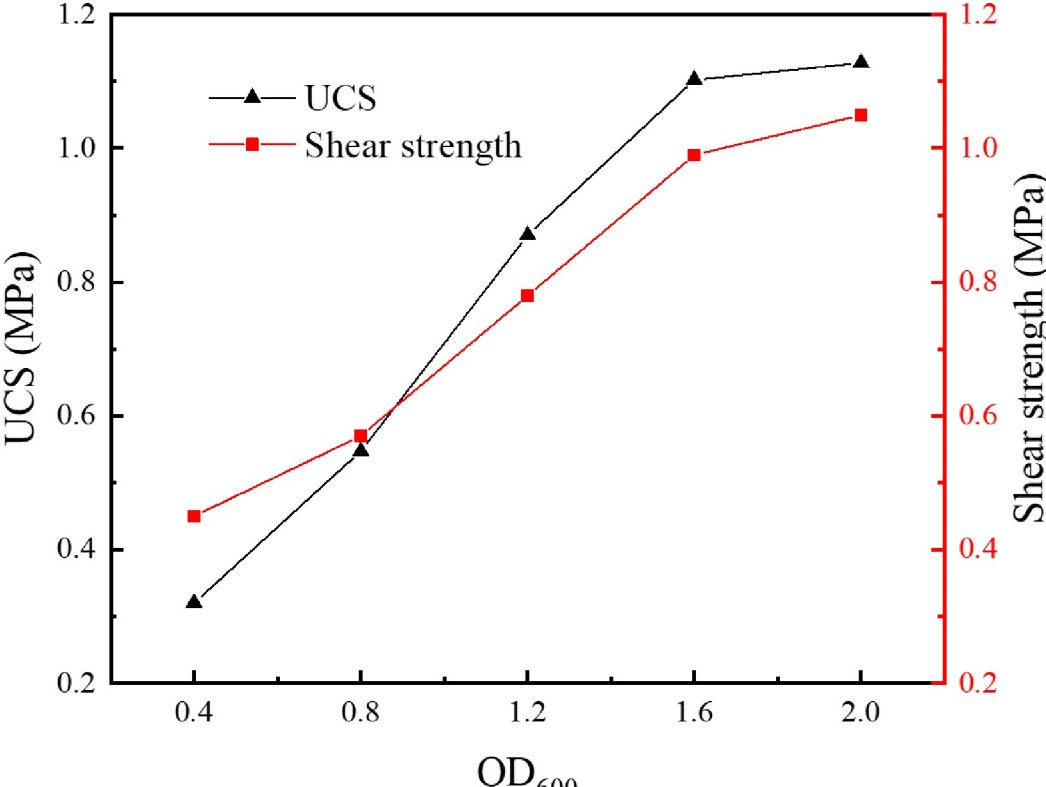

**Fig 7. Relationship between bacterial solution concentration and mechanical properties.**

studies, the spray repair method based on MICP technology is limited to repairing and coating the surface cracks of the specimen. The injection and percolation repair methods based on MICP technology are limited to the application in earthwork construction, and the process is relatively complicated and the cost is high. If it is applied to the curing of tailings, the cost is too high and the economic benefit is poor. In this paper, the tailings are cured by immersion curing. Compared with spraying, injection and percolation repair methods, immersion curing has the advantages of good effect, simple technology and easy practical application.

To study the combined effect of *S. pasteurii* and *C. denitrificans*, two mixing modes were adopted to mix the bacterial solution and tailings. For the first mode, the cultured *C. denitrificans* solution was centrifuged and the supernatant was taken out. The same volume of cultured *S. pasteurii* solution was poured in and dissolved to obtain a mixed solution containing two types of bacteria. The solution was then mixed evenly with the tailings and filled into the mold for specimen preparation. For the second mode, the two cultured bacterial solutions were mixed with the tailings respectively. The end of a 50 ml syringe with an inner diameter of 25 mm was cut off to make a cylindrical container and the length was marked (Fig 5A). The mixture of tailings and *C. denitrificans* solution was filled in the syringe and compacted (Fig 5B). The syringe was then put at the center of the steel mold and surrounded by the mixture of tailings and *S. pasteurii* solution (Fig 5C). The tailings inside the syringe was slowly injected into the steel mold and compacted. Finally, the specimen with tailings mixed with *C. denitrificans*

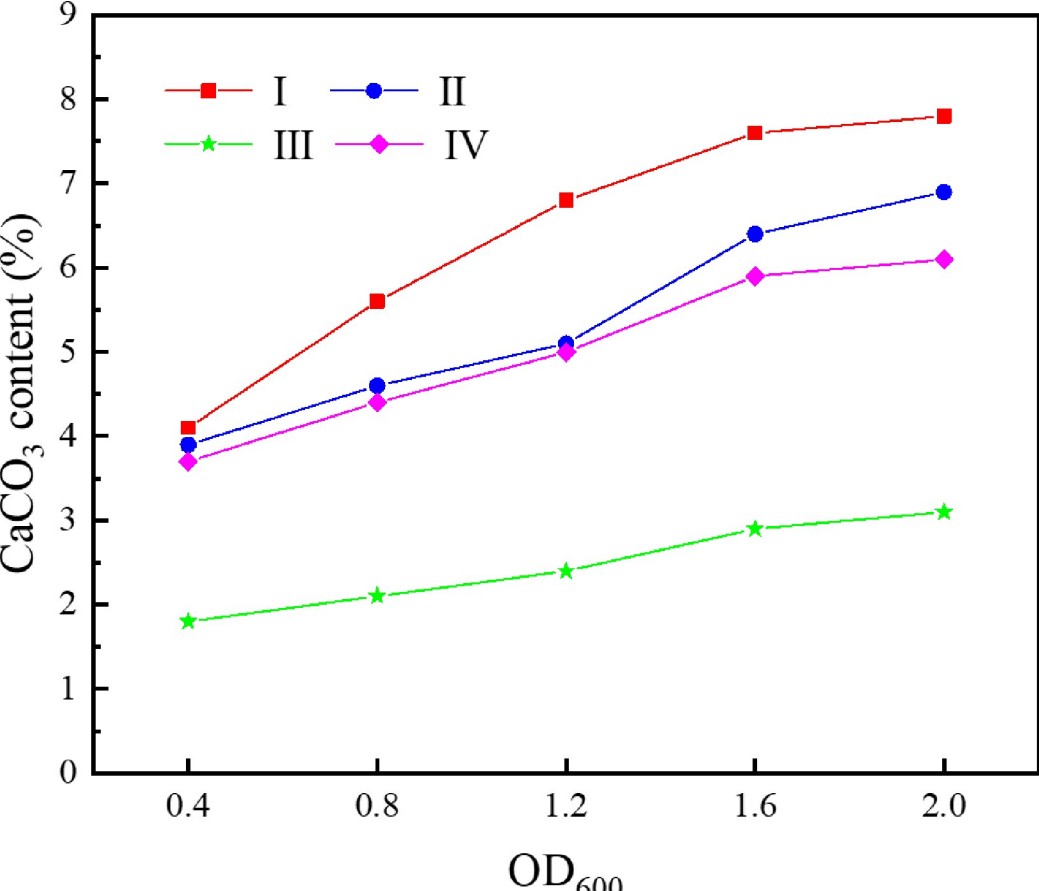

**Fig 8. Relationship between bacterial solution concentration and CaCO3 content in each region.**

in the center and *S. pasteurii* in the surrounding was obtained. After shaping, the specimen was wrapped and cured by the method as the same as that for the tailings specimen mixed with *S. pasteurii* solution.

### 3.2 Test on mechanical strength

Uniaxial compression tests and direct shear tests were conducted on the cemented tailings specimens. The device for uniaxial compression tests was Humboldt HM-5030 loading test machine manufactured in US. The precision of load sensor is 0.01N, the precision of displacement transducer is 0.001 mm and the maximum loading capacity is 50kN. The direct shear tests were carried out by YAW-3000 microcomputer-controlled electro-hydraulic servo rock mass direct shear test machine in Rock Mechanics Laboratory of Northeast University, China. A normal stress of 0.25 MPa was adopted. Cylindrical specimens with a dimension of $\phi 50$ mm*100 mm were tested. For each group of tests, the same tests were conducted on five specimens and the average values were taken for analysis. The mechanical parameters of tailings specimens under different conditions were obtained and the influences of various factors on the cementation effect were investigated.

### 3.3 SEM analysis

The SEM (Scanning Electron Microscope) analysis was conducted by the field emission scanning electron microscope in Analysis and Testing Center, Institute of Science and Technology, Northeast University, China. Model: Ultra Plus; Manufacturer: Zeiss, Germany; Resolution: 0.8 nm/15kv, 1.6nm/1kv; Accelerating voltage: 20V~30kV; Magnification: 12~1 million times; Secondary electron imaging; Fully digitized computer and manual parallel control system.

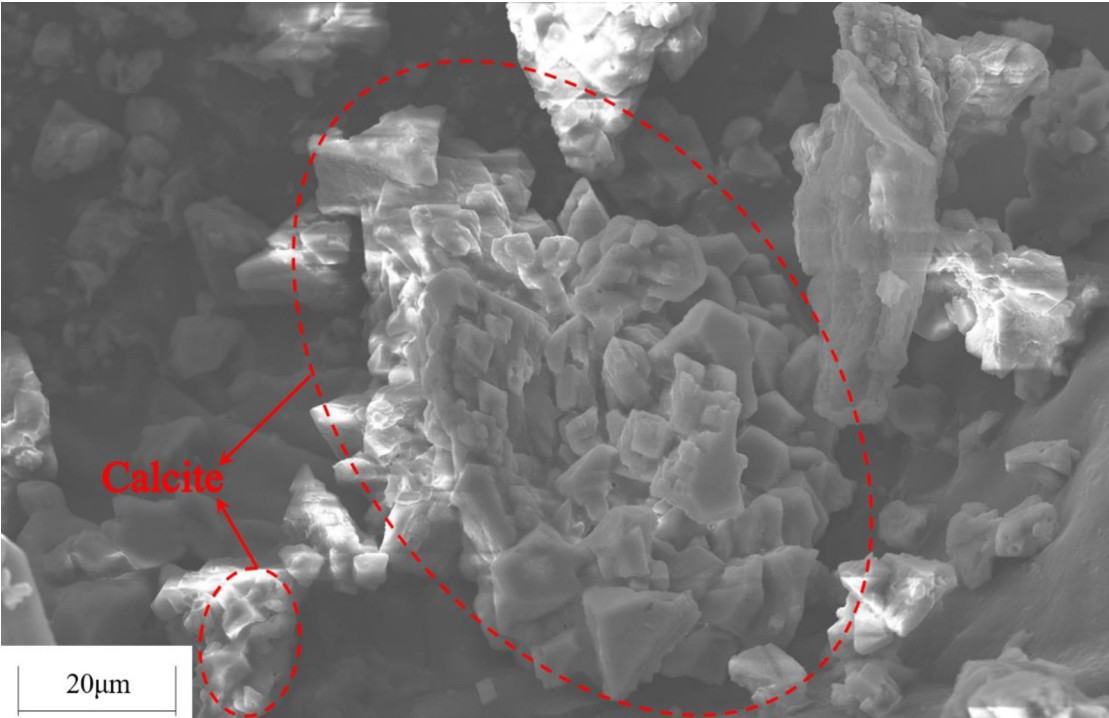

**Fig 9. SEM image of the tailings specimen for OD600 = 1.6.**

After sample preparation and conductive coating, the material structure and morphology formed in the MICP process were scanned and analyzed by SEM.

### 3.4 Test on calcium carbonate content

The $CaCO_3$ content in the cemented tailings specimen directly influences the cementation effect and the mechanical properties of the specimen. With appropriate adjustment based on the previous studies, the cemented tailings specimen was divided into 4 regions in order to analyze the uniformity of carbon carbonate precipitation, as shown in Fig 6. The different regions of the specimen were separated and pulverized with a mortar. The specimen powder was then dried in an oven. Its weight was recorded as $W_1$. The dry specimen powder was soaked and washed with deionized water and dried in an oven. The dry weight at this moment was recorded as $W_2$. The dry powder was then soaked and washed with 0.1mol/ml HCl solution until no bubbles were observed. The specimen powder was then washed with deionized water, filtered and dried. The weight at this moment was measured as $W_3$. The difference between $W_2$ and $W_3$ is the weight of $CaCO_3$ in the cemented specimen. The $CaCO_3$ content can then be calculated according to the original weight $W_1$, by using Eq (7). For each group of tests, 5 specimens were tested and the average values were taken.

$$w_{(CaCO_3)} = \frac{W_2 - W_3}{W_1}\%$$ 

(7)

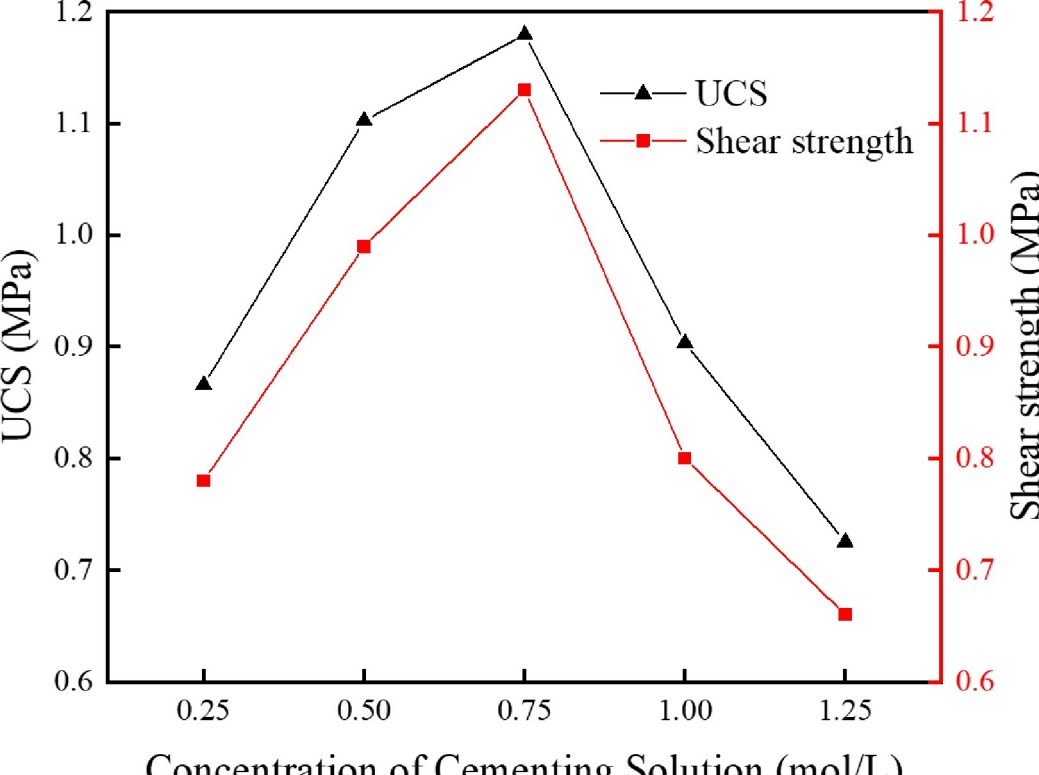

**Fig 10. Relationship between cementing solution concentration and mechanical properties.**

## 4. Results and discussions

### 4.1 Influence of bacterial concentration on cementation effect

As the optical density of bacterial solution is directly proportional to its concentration, the optical density corresponding to a wavelength of 600 μm (i.e., $OD_{600}$) was adopted to characterize the concentration of *S. pasteurii* solution. After being activated and cultured for 24h, the *S. pasteurii* solution was centrifuged and the supernatant was poured out. Based on the calculation results, different volumes of culture media were added to prepare bacterial solutions with $OD_{600}$ of 0.4, 0.8, 1.2, 1.6 and 2.0, respectively. The tailings were cemented by bacterial solutions with five different concentrations. In order to control the uniqueness of variables, the urea concentration in the cementing solution remained 0.5mol/L, and the original tailings without grinding were used. The tailings specimens were under immersion curing at a temperature of 30˚C for 28 days. The dried specimens were tested for uniaxial compressive strength and direct shear strength. The average values were taken, as shown in Fig 7. The $CaCO_3$ content was determined for each region and the average values were taken, as shown in Fig 8. The SEM image of tailings specimen corresponding to an $OD_{600}$ value of 1.6 is presented in Fig 9.

As can be seen from Fig 7, when $OD_{600}$ of the bacterial solution increases from 0.4 to 1.6, both the uniaxial compressive strength and the direct shear strength of the cemented tailings specimen increase significantly. The uniaxial compressive strength increases by 343.8% from 0.32MPa to 1.10MPa. The mechanical properties have been remarkably enhanced and can

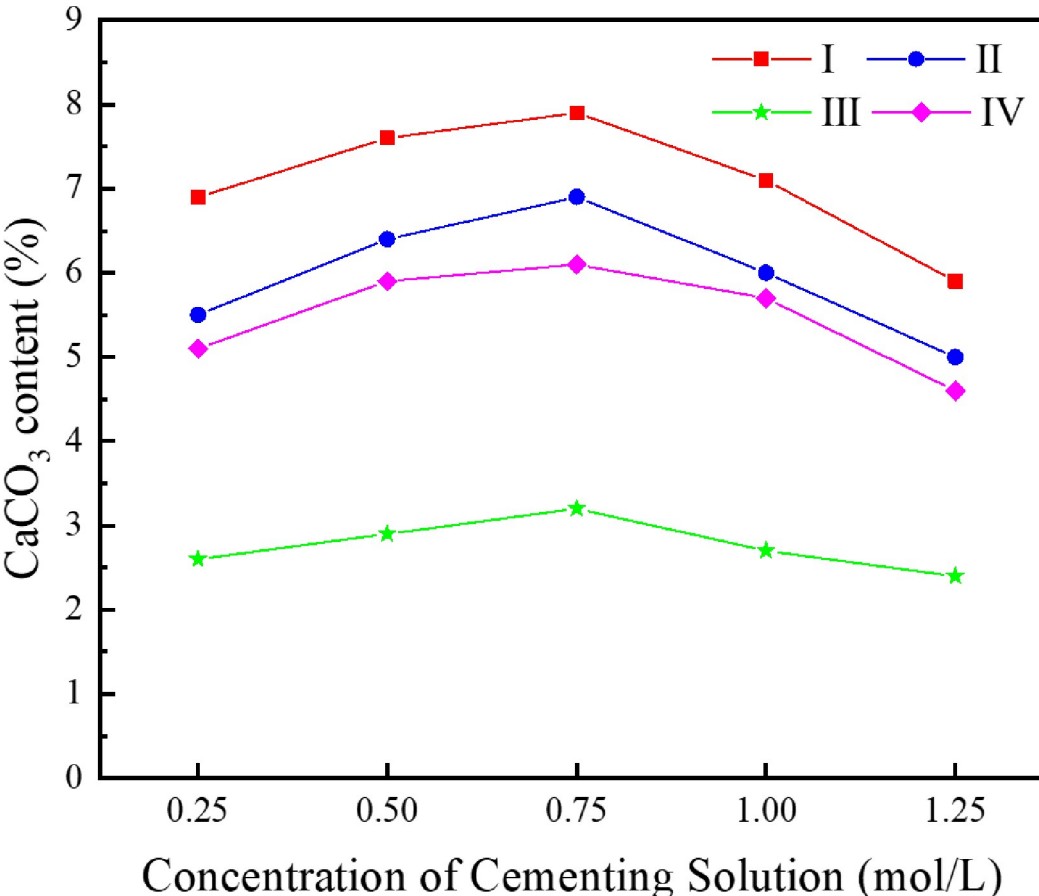

**Fig 11. Relationship between cementing solution concentration and CaCO3 content in each region.**

satisfy the strength requirement on mine backfill (the uniaxial strength of backfill has to be 0.7-2MPa for underground mining [22]. However, when $OD_{600}$ of the bacterial solution increases from 1.6 to 2.0, both the uniaxial compressive strength and the direct shear strength of the cemented tailings specimen vary slightly. It can be seen from Fig 8 that, except slight change in Region III, the $CaCO_3$ content in the other three regions increases significantly when $OD_{600}$ of the bacterial solution increases from 0.4 to 1.6, and changes slightly when $OD_{600}$ increases from 1.6 to 2.0. The cementation effect of *S. pasteurii* gradually increases with increasing bacterial solution concentration. However, when $OD_{600}$ exceeds 1.6, the growth and reproduction of *S. pasteurii* reach a certain balance and the increase of bacterial solution concentration has little impact on the number of bacteria in the specimen. As can be seen from Fig 9, a large amount of $CaCO_3$ is formed and closely bound in the cemented specimen, which greatly enhances the specimen strength. Therefore, when $OD_{600}$ exceeds 1.6, further increase of bacterial solution concentration cannot improve the concentration effect of microbes on the tailings. This part of the research proves that the MICP technology based on *S. pasteurii* can solidify the tailings into specimens with certain strength under the condition of immersion curing. It is confirmed that *S. pasteurii* has the potential to replace Portland cement as the cementing material of tailings. However, the urea hydrolysis of *S. pasteurii* will increase the emission of ammonia, which will also have a certain impact on the environment. In the follow-up, it can be considered to solve this problem by adjusting the concentration of reactants and adding ammonia gas absorbent.

## 4.2 Influence of cementing solution concentration on cementation effect

In order to investigate the influence of cementing solution concentration on the cementation effect of the MICP technique based on immersion curing, the cementing solutions with urea

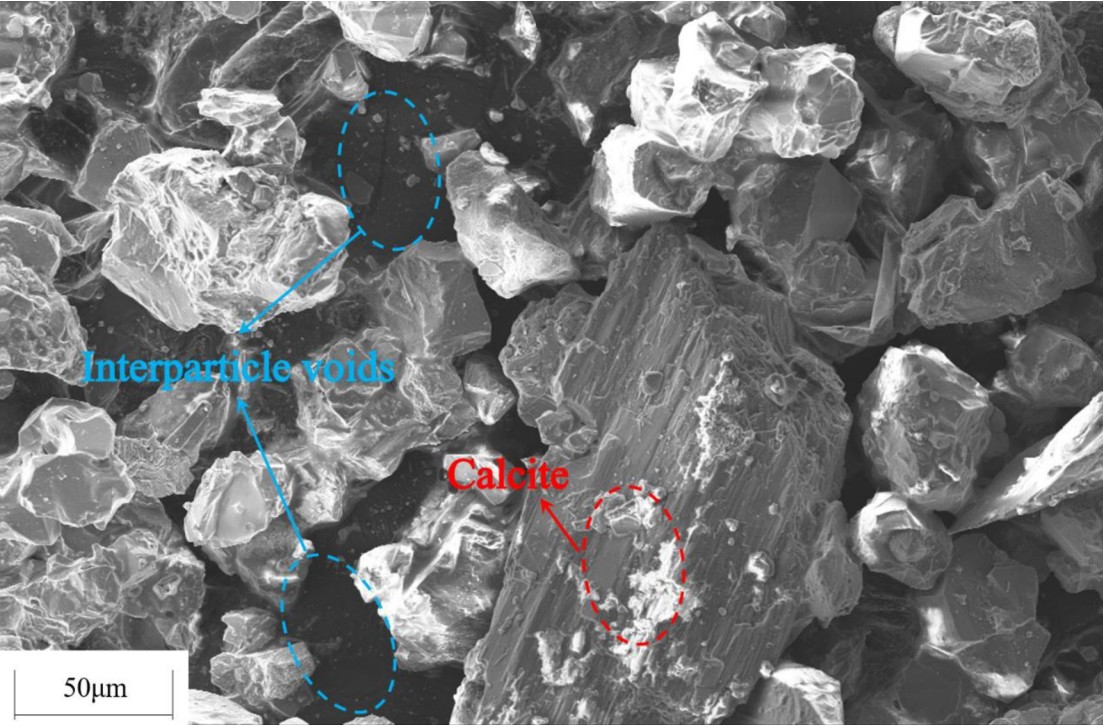

**Fig 12. SEM image of tailings specimen for cementing solution concentration of 0.25 mol/L.**

concentration of 0.25 mol/L, 0.50 mol/L, 0.75 mol/L, 1.00 mol/L and 1.25 mol/L, respectively, were prepared for tests. The *S. pasteurii* solution with an $OD_{600}$ value of 1.6 and the original tailings were mixed for sample preparation. The specimens were immersed in the cementing solutions with various urea concentration and cured at 30°C for 28d. The uniaxial compressive strength and direct shear strength were measured for dry specimens and the average values were taken, as shown in Fig 10. The $CaCO_3$ content in different regions were measured and calculated and the average values were taken, as shown in Fig 11. Fig 12 presents the SEM image of the specimen cemented in a cementing solution with urea concentration of 0.25mol/L.

As can be seen from Figs 10 and 11, with the increase of cementing solution concentration, the uniaxial compressive strength, the direct shear strength and the $CaCO_3$ content in each region of the cemented tailings specimen have similar trends. When the urea concentration in the cementing solution increases from 0.25mol/L to 0.75mol/L, the mechanical properties and the $CaCO_3$ content in each region of the cemented tailings specimen are enhanced gradually. The function of urea in the cementing solution is mainly to provide energy and raw materials for urea hydrolysis by *S. pasteurii*. When the concentration is below a certain level, the increase of urea concentration can promote the urea hydrolysis by *S. pasteurii*. As sufficient calcium ions are available in the cementing solution, the stronger the urea hydrolysis is, the more calcium carbonate precipitation is, and the stronger the mechanical properties of the cemented tailings specimen are. However, when the urea concentration in the cementing solution increases from 0.75 mol/L to 1.25 mol/L, the $CaCO_3$ content in each region decreases slightly, and the uniaxial compressive strength and the direct shear strength of specimen decreases rapidly. It can be seen from Fig 12 that, when the urea concentration is 0.25 mol/L, although a few

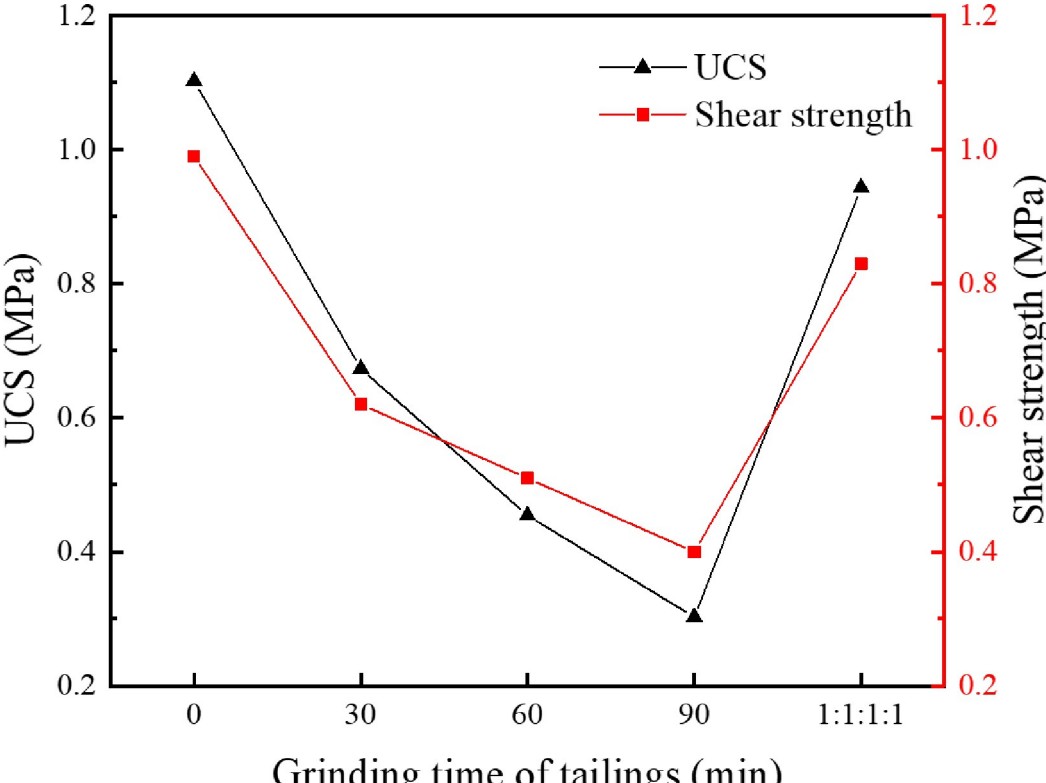

**Fig 13. Relationship between tailings particle size and mechanical properties.**

$CaCO_3$ particles are formed on the tailings particles, they are not bound as a whole and cannot fill up the gaps between tailings particles. Hence, the specimen strength is relatively low. When the urea concentration in the cementing solution is higher than 0.75 mol/L, the increase of urea concentration can no longer promote, instead restrains the urea hydrolysis by *S. pasteurii*. The main reason is that too high urea concentration leads to an environment unsuitable for the growth and reproduction of *S. pasteurii*. Consequently, the urea hydrolysis process is slowed down, less $CaCO_3$ is formed in the specimen, and the mechanical properties are weakened.

### 4.3 Influence of tailings particle size on cementation effect

In order to investigate the influence of tailings particle size on the cementation effect, the original tailings were ground by a vertical planetary ball mill for 30 min, 60 min and 90 min, respectively, and the original tailings were mixed with the three types of ground tailings in a mass ratio of 1:1:1:1 to prepare a tailings material with a new particle size distribution. Five types of specimens, i.e., the original tailings (ground for 0min), the tailings ground for 30 min, 60 min, 90 min, and the mixture in a mass ratio of 1:1:1:1, with the same composition and different particle size distributions were obtained. The *S. pasteurii* solution with an $OD_{600}$ value of 1.6 was mixed with the tailings having various particle size distributions to prepare the tailings specimens. The specimens were immersed in the cementing solution with a urea

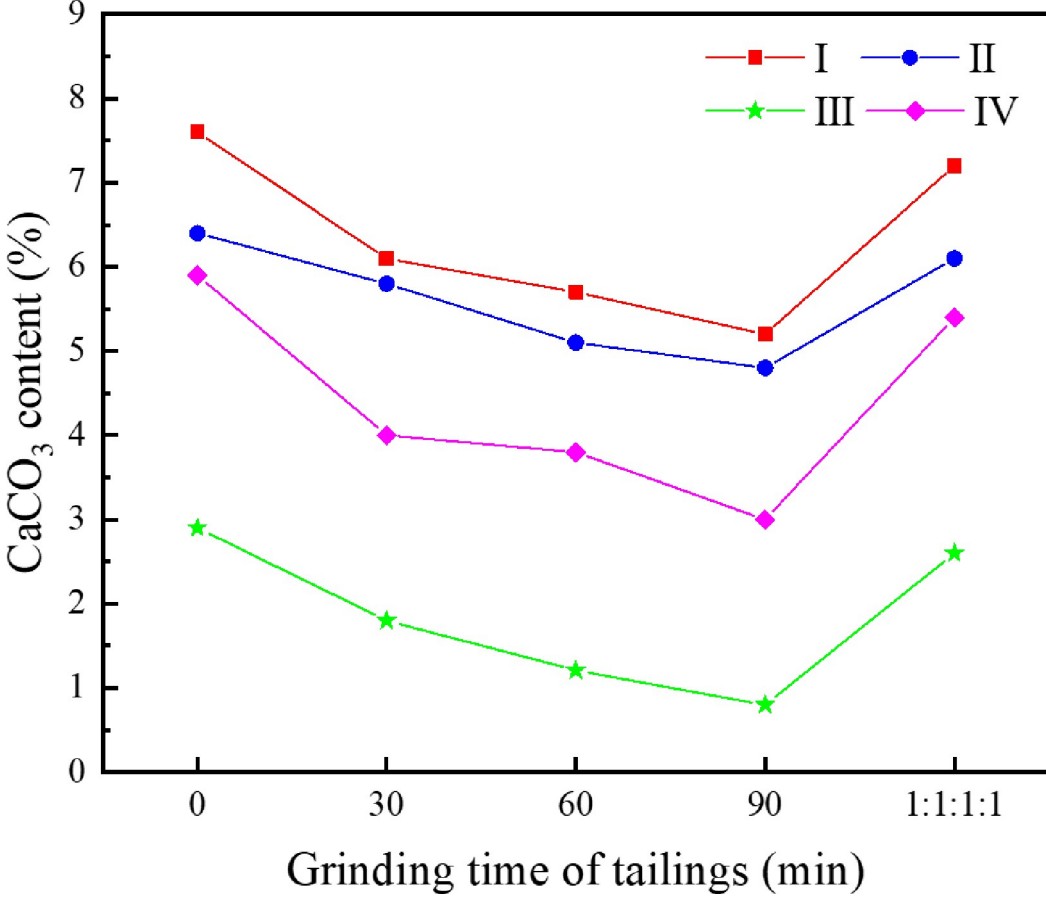

**Fig 14. Relationship between tailings particle size and CaCO3 content in different regions.**

concentration of 0.5mol/L at 30˚C for 28 d. Tests were conducted on the dry specimens for uniaxial compressive strength and direct shear strength. The average values were taken, as shown in Fig 13. The $CaCO_3$ content in each region was measured and calculated. The average values were taken and shown in Figs 14 and 15 presents the SEM image of the cemented tailings specimen which was ground for 90 min.

As can be seen from Figs 13 and 14, among the tailings specimens with five types of particle size distributions, the cementation effect is the best for the original tailings. The compressive strength and the direct shear strength of the original tailings after cementation are much higher than those of ground specimens. The reasons are: for immersion curing, the cementing solution can easily pass larger tailings particles and enter the interior of the specimen, and participate in the urea hydrolysis process of *S. pasteurii*, leading to precipitation of $CaCO_3$. When the grinding time is longer, the proportion of small particles in the tailings is higher, it is harder for the cementing solution to enter the specimen and participate in the urea hydrolysis process. In addition, the smaller the particle size is, the fewer the pores between the particles are. *S. pasteurii* is a kind of aerobic bacteria, which can only survive in the presence of oxygen. When there are fewer pores, most *S. pasteurii* would die and cannot hydrolyze urea due to hypoxia. Hence, the cementation effect is poor. It can be seen from Fig 15, when the proportion of small or medium-size particles is too high, little $CaCO_3$ is formed in the pores between small particles and hence cannot bind fine particles. Therefore, the strength of the cemented tailings which was ground for 90 min is relatively low. Among the tailings specimens with different particle size distributions (except the original tailings), the cementation effect for the specimen with a mass ratio of 1:1:1:1 is the best, as the existence of larger particles in the specimen with a mass ratio of 1:1:1:1 provides support and enough space for the survival and growth of *S. pasteurii*. Furthermore, the small particles act as fine aggregates between pores. Hence, the coarse

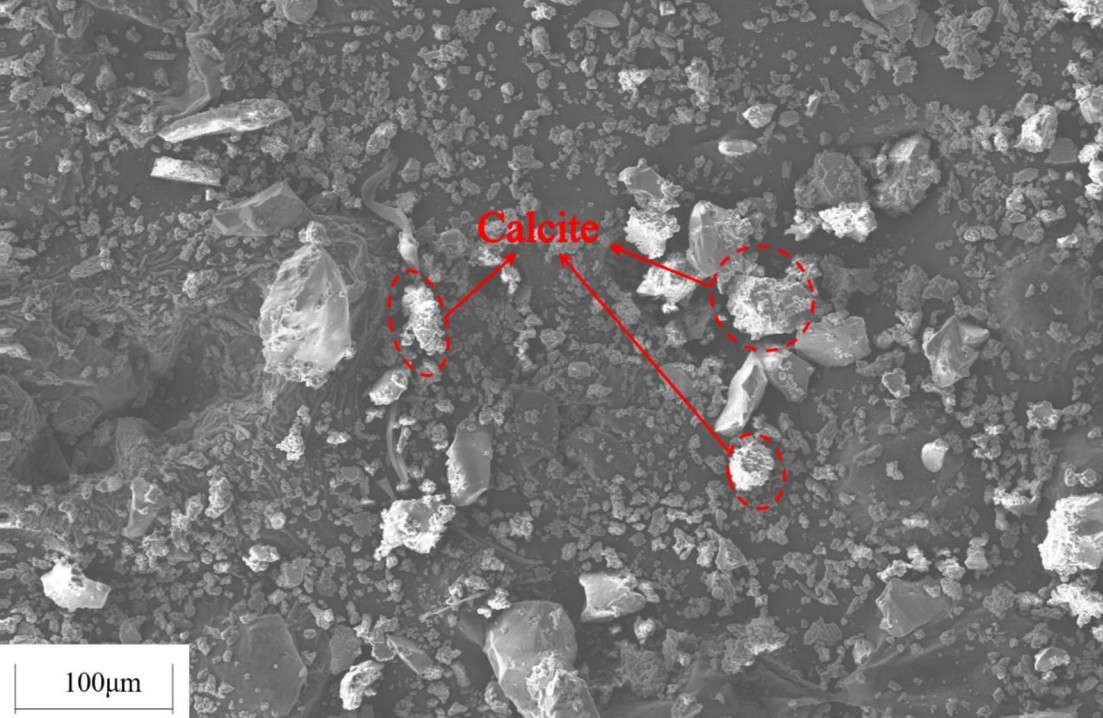

**Fig 15. SEM image of cemented tailings specimen.**

and fine particles can still be cemented when the $CaCO_3$ crystals formed by the MICP process are small and the cementation effect is good.

## 4.4 Influence of curing temperature on cementation effect

When studying the influence of temperature on the cementation effect, the *S. pasteurii* solution with $OD_{600}$ of 1.6 was mixed with the original tailings and the specimens were cultured in a cementing solution with urea concentration of 0.5 mol/L. The specimens were under immersion curing at 20˚C, 25˚C, 30˚C, 35˚C and 40˚C, respectively, for 28 d. The uniaxial compressive strength and the direct shear strength were tested for dried specimens and the results are shown in Fig 16. The $CaCO_3$ content in each region was measured and calculated, as presented in Fig 17.

According to the curves shown in Fig 16, the uniaxial compressive strength and the shear strength of the cemented specimen reach the peak values at a curing temperature of 30˚C, indicating that 30˚C is the optimal temperature for urea hydrolysis by *S. pasteurii* and precipitation of $CaCO_3$. 30˚C is also the most suitable temperature for culturing of *S. pasteurii*, which has high activity and can grow and reproduce rapidly at this temperature. When the temperature exceeds 30˚C, the mechanical properties of the cemented specimen weaken rapidly with increasing temperature. High temperature is unfavorable for the survival of *S. pasteurii*. Most *S. pasteurii* would die and cannot hydrolyze urea any more.

According to the curves shown in Fig 17, the variation trend of $CaCO_3$ content in Regions I, II and III is similar to that of mechanical properties of the cemented specimen, i.e., reaching the peak at a temperature of 30˚C, being less at a temperature higher or lower than 30˚C. However, the temperature has little impact on the $CaCO_3$ content in Region III, which is in the

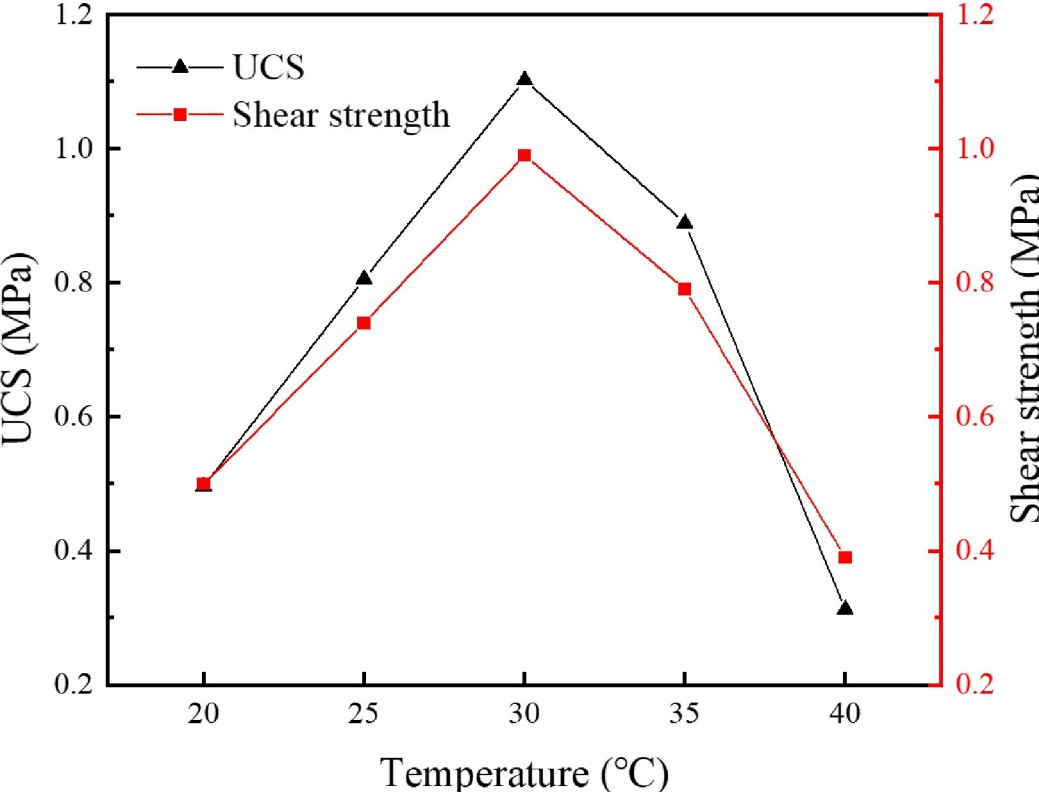

**Fig 16. Relationship between curing temperature and mechanical properties.**

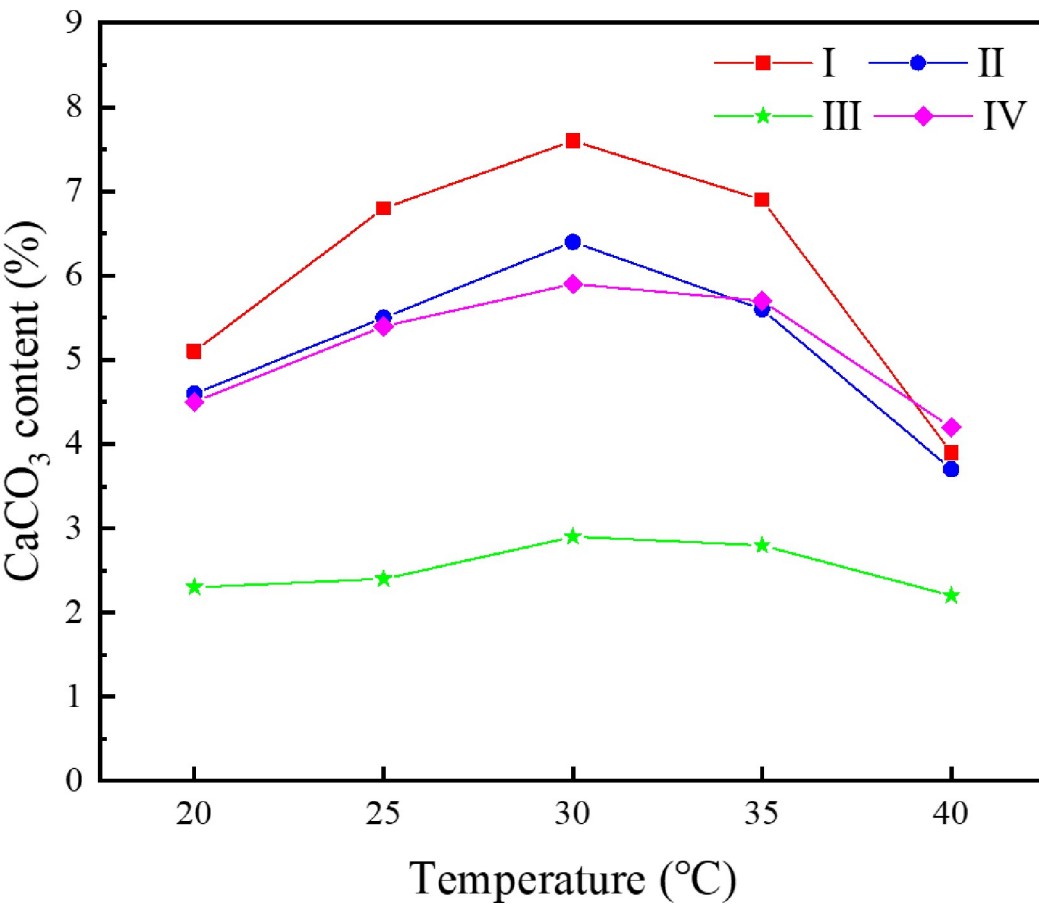

**Fig 17. Relationship between curing temperature and CaCO3 content in each region.**

center of the specimen and less affected by the curing temperature. In Region III, before the temperature decreases or increases to an value unfavorable for *S. pasteurii*, a certain amount of $CaCO_3$ has already formed in the center. As can be seen from Fig 18, a great amount of $CaCO_3$ is formed in the interior of the specimen cured at 30˚C. The pores between tailings particles are well filled and the cementation effect is good. Hence, the strength of the specimen cured at 30˚C is relatively high.

## 4.5 Distribution of products

According to the above results, the variation trends of the uniaxial compressive strength and the direct shear strength of cemented tailings specimens are similar to that of $CaCO_3$ content in each region. However, the $CaCO_3$ content is obviously different in various regions. The $CaCO_3$ precipitation in each region of the cemented specimen under various test conditions is summarized and presented in Fig 19. It can be seen that the $CaCO_3$ content in the top (Region I) and side (Region II) of the cemented tailings specimens subjected to immersion curing is higher, followed by the bottom (Region IV) and the interior (Region II). As the *S. pasteurii* is a kind of aerobic bacteria, sufficient amount of oxygen is required for its urea hydrolysis process and the pH value of the environment is increased, leading to binding of calcium ions to carbon dioxide and precipitation of calcium carbonate precipitation. Although an air pump is connected to the test device, oxygen can hardly enter the interior of the specimen due to few pores

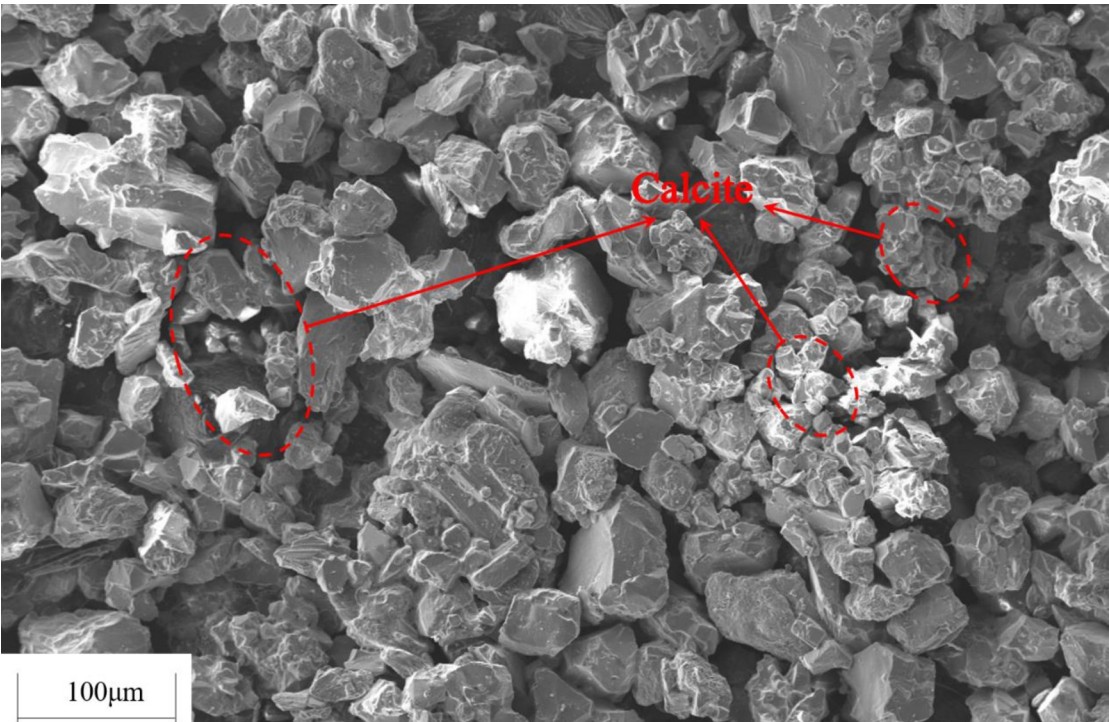

**Fig 18. SEM image of tailings specimen after cementation at 30˚C.**

between tailings particles. Therefore, the microbes in the top and side of the specimen can contact with more oxygen, and more calcium carbonate is formed in these regions. Some microbes in the interior of the specimen die due to hypoxia and can no longer hydrolyze urea. Although the bracket is perforated, the holes are gradually blocked with the accumulation of calcium carbonate. Hence, the $CaCO_3$ content at the bottom of the specimen is lower compared to that at the top. In addition, it is difficult for the cementing solution to enter the interior through tailings particles, resulting in the lack of urea and calcium ions and thus less calcium carbonate precipitation in the interior.

## 4.6 Comparison of cementation effect under different bacteria mixing modes

In order to solve the uneven distribution of $CaCO_3$ in each region of the specimen cemented by the MICP technique with aerobic bacteria, the urea hydrolysis by *S. pasteurii* and the denitrification by *C. denitrificans* were combined. Two mixing modes, i.e., direct mixing and mixing by using a syringe, were adopted to mix the bacterial solution and the tailings. The mixed specimens were immersed and cured under the same conditions. The uniaxial compressive strength, the direct shear strength and the $CaCO_3$ content in each region were measured for dried specimens, as shown in Fig 20.

As can be seen from Fig 20, under the same condition, the average $CaCO_3$ content in Region III is 2.9% when only *S. pasteurii* is used. The $CaCO_3$ content in Region III of the specimen prepared by the direct mixing mode (Mode I) is 5.1%, which is increased by 75.9%. The $CaCO_3$ content in Region III of the specimen mixed by a syringe (Mode II) is 6.5%, which is increased by 124.1%. When both *S. pasteurii* and *C. denitrificans* are used for tailings cementation, the $CaCO_3$ content in Region III is greatly enhanced by either of the two mixing modes.

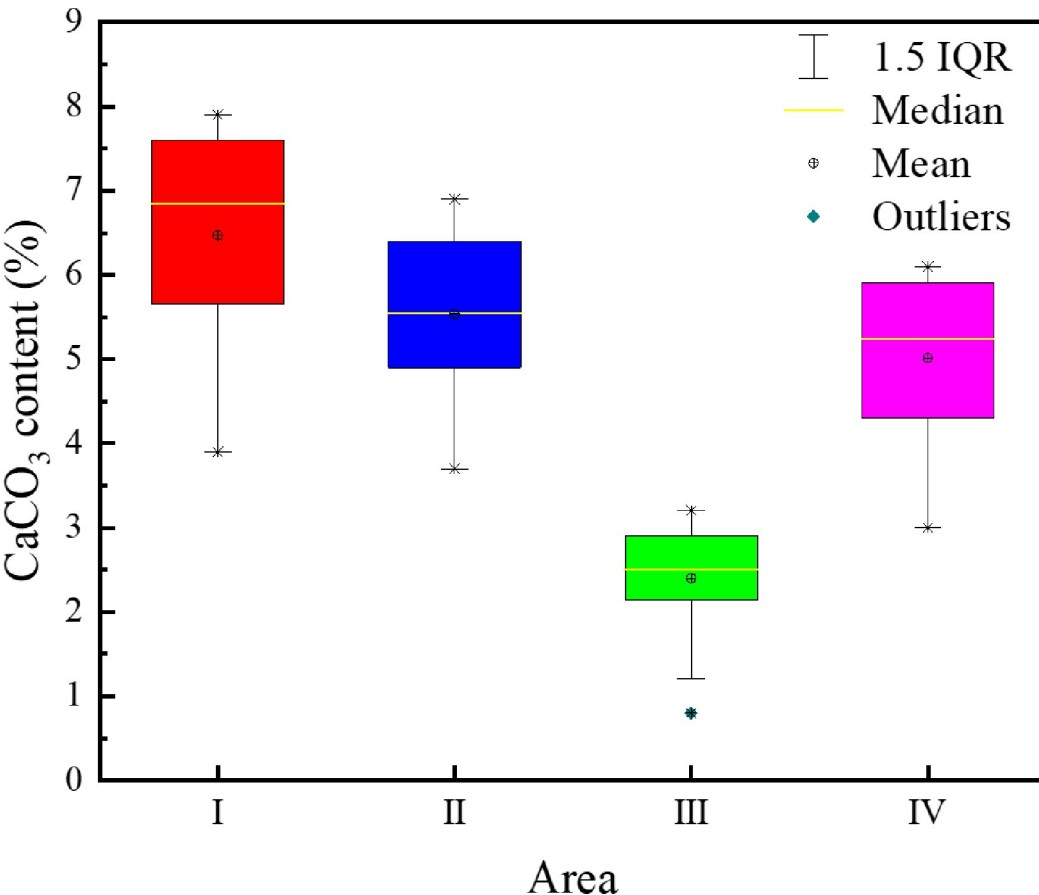

**Fig 19. CaCO3 precipitation in different regions.**

It indicates that denitrification of facultative anaerobes is dominant in the anoxic environment inside the specimen. The $CaCO_3$ distribution in various regions is relatively uniform and more $CaCO_3$ is precipitated inside the specimen mixed by Mode II. As both *S. pasteurii* and *C. denitrificans* exist in the interior of the specimen mixed by Mode I, *S. pasteurii* cannot hydrolyze urea due to hypoxia, but it competes for living space and nutrient with *C. denitrificans*. As a result, some *C. denitrificans* cannot carry out the denitrification process normally. However, in the specimen mixed by Mode II, only *C. denitrificans* exist in the interior, hence more $CaCO_3$ is formed. Although the $CaCO_3$ content increases in the specimen with mixed bacteria, the formation rate of $CaCO_3$ by denitrification is slower compared to that by urea hydrolysis. The amount of $CaCO_3$ due to denitrification is only 62% of urea hydrolysis, and cannot reach the $CaCO_3$ content at the top and side of the specimen. As the $CaCO_3$ content increases slightly in the specimen with mixed bacteria, it has little impact on the overall mechanical properties of the specimen. The uniaxial compressive strength and the direct shear strength of the specimen cemented by mixed bacteria increases by 9.3% and 14.3%, compared with the specimen cemented by *S. pasteurii* only.

## 5. Conclusions

In this paper, experimental studies were carried out on the influencing factors for cementing tailings specimens by the MICP technique with *S. pasteurii* under immersion curing. The

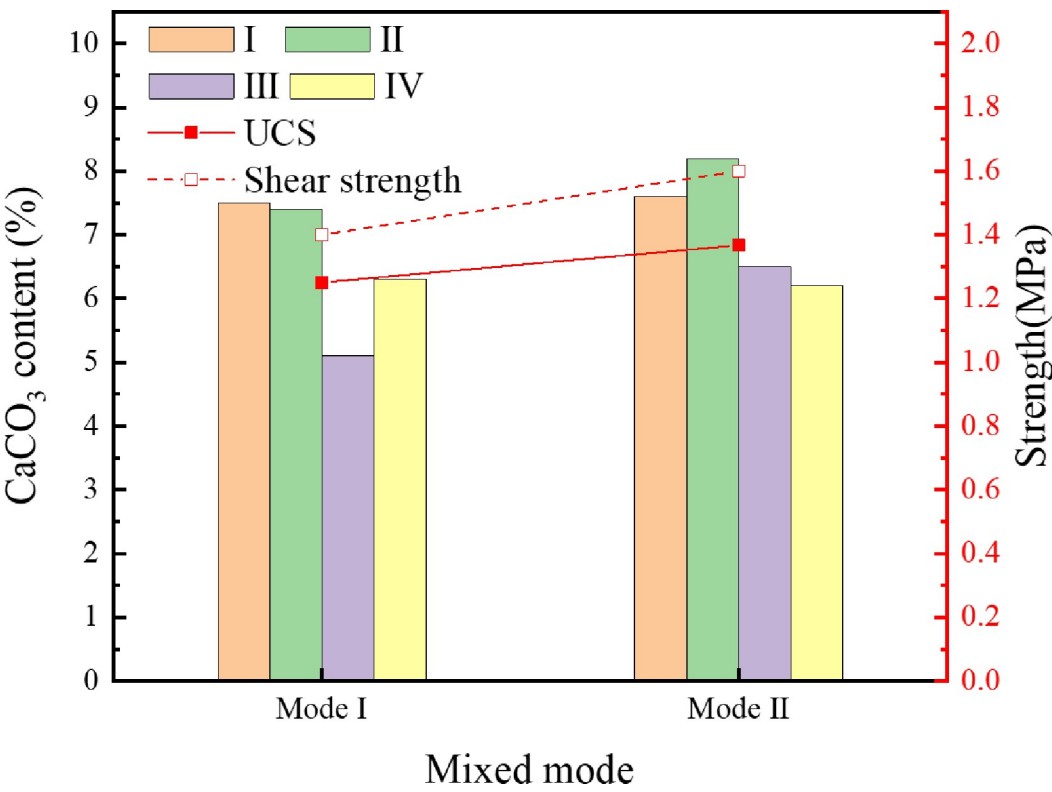

**Fig 20. Cementation effect of different mixing modes.**

single variable method was adopted to investigate the influences of bacterial solution concentration, cementing solution concentration, tailings particle size and temperature on the cementation effect. The following conclusions can be drawn:

1. With the increase of bacterial solution concentration, the uniaxial compressive strength and the direct shear strength increased first and then gradually decreased. When $OD_{600}$ of the *S. pasteurii* solution was 1.6, the cementation effect was no longer enhanced rapidly with increasing bacterial solution concentration. High urea concentration constrained the growth of *S. pasteurii* to a certain extent. When the urea concentration was 0.75 mol/L, the cementation effect was optimal. With the increase of temperature, the uniaxial compressive strength and the direct shear strength tended to increase first and then decrease. The cementation effect was optimal at a temperature of 30˚C.

2. The $CaCO_3$ content in each region of the specimen varied in a trend similar to that of uniaxial compressive strength and direct shear strength. The $CaCO_3$ content was positively correlated with the mechanical properties of the specimen.

3. Two mixing modes were adopted to mix *S. pasteurii* and *C. denitrificans* with the tailings. The test results indicated that combined action of aerobic bacteria and facultative anaerobes can improve the amount of $CaCO_3$ precipitation and result in more uniform distribution of $CaCO_3$ in the interior of the specimen. In addition, the cementation effect in the specimen mixed by a syringe was improved more obviously.

## Supporting information

**S1 Data.**

(XLSX)

## Author Contributions

**Conceptualization:** Changyu Jin.

**Data curation:** Changyu Jin.

**Formal analysis:** Changyu Jin.

**Methodology:** Huiyang Liu.

**Writing – original draft:** Mingxiao Guo, Yunfeng Wang, Jinyao Zhu.

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
