## [Decision Letter · Decision Letter 0]

16 May 2022

PONE-D-21-40689Experimental study on tailings cementation by MICP technique with immersion curingPLOS ONE

Dear Dr. Changyu Jin,

Thank you for submitting your manuscript to PLOS ONE. After careful consideration, we feel that it has merit but does not fully meet PLOS ONE’s publication criteria as it currently stands. Therefore, we invite you to submit a revised version of the manuscript that addresses the points raised during the review process.

ACADEMIC EDITOR: Please carefully check the writing before re-submission.==============================

We look forward to receiving your revised manuscript.

Kind regards,

Kedsarin Pimraksa, PhD

Academic Editor

PLOS ONE

**Journal requirements:**

“This work was supported by the Fundamental Research Funds for Central Universities, China (N2101041).”

5. PLOS requires an ORCID iD for the corresponding author in Editorial Manager on papers submitted after December 6th, 2016. Please ensure that you have an ORCID iD and that it is validated in Editorial Manager. To do this, go to ‘Update my Information’ (in the upper left-hand corner of the main menu), and click on the Fetch/Validate link next to the ORCID field. This will take you to the ORCID site and allow you to create a new iD or authenticate a pre-existing iD in Editorial Manager. Please see the following video for instructions on linking an ORCID iD to your Editorial Manager account: https://www.youtube.com/watch?v=_xcclfuvtxQ.

**Reviewers' comments:**

Reviewer's Responses to Questions

**Comments to the Author**

1. Is the manuscript technically sound, and do the data support the conclusions?

Reviewer #1: Partly

Reviewer #2: Yes

2. Has the statistical analysis been performed appropriately and rigorously? 

Reviewer #1: No

Reviewer #2: Yes

3. Have the authors made all data underlying the findings in their manuscript fully available?

Reviewer #1: No

Reviewer #2: Yes

4. Is the manuscript presented in an intelligible fashion and written in standard English?

Reviewer #1: Yes

Reviewer #2: Yes

5. Review Comments to the Author

Reviewer #1: In the manuscript titled „Experimental study on tailings cementation by MICP technique with immersion curing” experimental study on MICP performance in the cementation of tailing under different condition are presented. Some experimental findings regarding the influences of bacterial concentration, concentration of urea solution, particle size, temperature and mixture mode of bacteria were addressed.

Although the reviewer think the most results are reasonable and straightforward, as many qualitative as well as quantitative investigations on the different influence factors on MICP treatment are already available in the current literature, compared to other researches the study in this manuscript seem to be not deep enough. There is lack of quantification and comprehensive analysis of the experimental observations. Thus, it’s hard to get more insight into MICP cementation of tailing based on this study.

Further, the reviewer questions also the novelty and scientific contributions of the present study. On one side, usually, one of main contributions of the experimental study is that the experiments can provide data base which can be used as reference for other experimental study as well as for validation the numerical study. However, in the manuscript only few data points are available.

On the other side, only some qualitative conclusions are addressed in the manuscript, and the tests were carried out on specific tailing material in laboratory scale. Therefore the experimental findings cannot be applied to other further studies.

Except for the main concerns above, the design of some tests is so far not clear to me (please see the specific comments below). Moreover, the language aspect and format should be checked (some suggestions with regard to the language and format aspects can also be found in the specific comments below).

Specific comments:

• Line 15: what does “ore energy” mean? Does it mean ore and energy?

• Line 36: There should be always a space between worlds and bracket. e.g. should be “underground mining (Li et al. 2021; Wang et al. 2018)” instead of “underground mining(Li et al. 2021; Wang et al. 2018)”

• Lines 45-47: the expression sounds a bit weird. It should be rephrased

• Lines 54-57: The description of urea hydrolysis is improper. Urea hydrolysis actually refer to the reaction process, in which urea is consumed, and carbonate ion and NH4 are produced. The authors should also mention that in the MICP treatment expect for urea and bacteria, Ca2+ should be additionally provided in the system. For better explanation of the chemical reactions by MICP, one should give the chemical equation here.

• Section 4.1: Any explanation for the experimental observation that after excess 1.6 OD, no significant increase of calcite production was observed? Actually this observation is quite common. With the increase of bacterial amount, the urease activity is improved. However, after reaching a certain bacterial mass competition among the bacteria increases due to the insufficient nutrients which on the contrary leads to the increase of bacterial decay thus decreases the activity. This can usually be seen in the growth rate of biomass. By means of changing the ambient environment of bacteria e.g. changing the nutrition receipts, or providing more nutrition) the bacteria activity can be further improved. Considering this, it is more interesting to investigate the optimal ratio of the bacterial concentration with respect to the urea/calcium concentration rather than to study these two concentrations separately.

• Section 4.2: not sure if the conclusions drawing here make sense. Firstly, this observation is of no general applicability. Second, if one want to establish the relationship between urea concentration and bacterial urease activity, it is better to analyze the urease activity of bacteria in cases of different combination of bacteria concentration and urea concentration based on monitoring the decrease rate of urea mass during the test (e.g see. Xiao et al., 2020).

• Section 4.3: Only uniform and homogenous distribution of different particle sizes is investigated. However, in the in-situ MICP cementation of tailing, the heterogeneous particle size distribution in a large ´field is expected.

• Section 4.4: To analyze the relationship between temperature and spatial distribution of produced calcite, one should look into the urease activity of bacteria at different temperature (see e.g. Xiao et al., 2020).

Reference:

Xiao, Y., Wang, Y., Wang, S., Evans, T. M., Stuedlein, A. W., Chu, J., … Liu, H. (2021). Homogeneity and mechanical behaviors of sands improved by a temperature-controlled one-phase MICP method. Acta Geotechnica, 4.

Reviewer #2: The authors have presented a work performed to stabilize the tailing by MICP. The work is original and can be recommended for publication IN PLOS ONE. However, the following comments must be carefully addressed.

1. The authors should be careful when writing the scientific names of bacteria species. The species name should be in italic - revise throughout the manuscript.

2. There should be a space between number and units. For example, 70mm should be written as 70 mm.

3. I wonder, what is the new finding in the manuscript, comparing to the published article? If it is optimization, that has already been published in previous works. Authors should highlight this in the latter part of the introduction section.

4. When writing OD600, 600 should be subscripted.

5. What is the advantage in immersion curing, compared to the spraying/ injection/ percolation? How the proposed method is suitable for tailing applications? The discussion should be expanded in a way to answer these questions.

5. Similar to Ordinary Portland cement, the use of MICP also has adverse effect related to the ammonia emission/ release. The authors should briefly discuss the negative side of MICP technique, and should propose the possible methods to overcome this issue. Several recently published works confirmed that the ammonium by-products can be managed by struvite precipitation (https://doi.org/10.1007/s13762-021-03138-z), zeolite use (https://doi.org/10.1520/GTJ20170353), and etc. The authors are recommended to refer those papers and expand the discussion.

6. I still confused about one thing. Can this be practical to implement MICP to tailing? In the MICP, the bacteria as well as resources should reach the depths to got treated at those levels. However, the tailing materials are more likely to be very fine (in the range of nano scale). Will it be possible to penetrate the bacteria? Moreover, tailing used to have organic content. A recent study showed only a limited improvement is possible in fine and organic soil materials. Refer the following paper (https://doi.org/10.3389/fenvs.2021.690376) and enhance the discussion, which helps to improve the manuscript standard.

7. Regarding the concentration of cementation solutions, 0.25mol/L, 0.50mol/L, 0.75mol/L, 1.00mol/L and 1.25mol/L, what is the demerit in using 1 mol/L or 1.25 mol/L? Few previous works recommend the use of 1 mol/L. Scientific explanation may be needed.

8. In some places, 3 in CaCO3 is not subscripted. Revision needed throughout.

9. Is this the first article using Castellaniella denitrificans?

6. PLOS authors have the option to publish the peer review history of their article (what does this mean?). If published, this will include your full peer review and any attached files.

Reviewer #1: No

Reviewer #2: No

---

## [Author Response · Author response to Decision Letter 0]

29 Jun 2022

Responses to reviewer comments have been submitted in file format.

---

## [Editor Report · Decision Letter 1]

18 Jul 2022

Experimental study on tailings cementation by MICP technique with immersion curing

PONE-D-21-40689R1

Dear Dr. Changyu Jin

We’re pleased to inform you that your manuscript has been judged scientifically suitable for publication and will be formally accepted for publication once it meets all outstanding technical requirements.

Kind regards,

Kedsarin Pimraksa, PhD

Academic Editor

PLOS ONE
---

## [Editor Report · Acceptance letter]

21 Jul 2022

PONE-D-21-40689R1 

Experimental study on tailings cementation by MICP technique with immersion curing 

Dear Dr. Jin:

I'm pleased to inform you that your manuscript has been deemed suitable for publication in PLOS ONE. Congratulations! Your manuscript is now with our production department. 

Kind regards, 

on behalf of

Dr. Kedsarin Pimraksa 

Academic Editor

PLOS ONE